# OpenReview forum: "Open Data Synthesis for Deep Research"
_ICLR.cc/2026/Conference — ICLR 2026 Poster_

### Official Review · Reviewer_NXXo · 2025-10-29

**Soundness:** 3
**Presentation:** 3
**Contribution:** 4
**Rating:** 6
**Confidence:** 5

**Summary:**

This paper introduces **InfoSeek**, a novel data synthesis framework that formulates agentic search as a **Hierarchical Constraint Satisfaction Problem (HCSP)**. Through a **Diffusion–Retrospection** process, InfoSeek constructs over **50K QA pairs** and **16.5K reasoning trajectories**, achieving substantial improvements across multiple benchmarks, including **BrowseComp-Plus**, thereby validating the approach’s effectiveness.

**Strengths:**

* Clear and principled theoretical formulation via HCSP (Eqs. (1)–(3)).
* Well-designed synthesis algorithm producing layered, verifiable data (Algorithm 1).
* Large, high-quality dataset (Table 2) with controlled complexity.
* Strong experimental results: InfoSeeker-3B achieves **15.3%** on BrowseComp-Plus, outperforming GPT-4.1 and Search-R1-32B.
* Open-sourced code and data ensure transparency and reproducibility.

**Weaknesses:**

* **Limited baseline details:** RL-based baselines (e.g., Search-R1, InForage) lack hyperparameter disclosure.
  → *Recommendation:* Provide full training details for reproducibility.
* **Insufficient failure analysis:** While accuracy improvements are notable, deeper case-level error analysis is needed.
  → *Recommendation:* Include failure taxonomy or qualitative examples.
* **Heuristic ambiguity:** The implementation of “blur parent” and “depth expansion” lacks concrete examples.
  → *Recommendation:* Add illustrative instances in the appendix.

**Questions:**

* Can the HCSP formulation be generalized to multimodal or retrieval-augmented settings?
* Did the authors explore continuous reward functions (e.g., confidence-weighted accuracy) during RL training?

---

> ### Author Response · Authors · 2025-11-21
>
> We sincerely thank reviewer **NXXo** for the very positive assessment of our work, including the strong rating on contribution and the recognition of our theoretical formulation, the design of Algorithm 1, the quality of our HCSP-structured dataset, and the substantial performance gains over prior systems such as GPT-4.1 and Search-R1-32B. Your thoughtful comments are greatly appreciated. We address them point-by-point below.
>
>
>
> ------
>
>
>
>
>
> ## **1. Baseline Details and Reproducibility**
>
>
>
>
>
> Thank you for highlighting the need for clearer baseline specifications. All RL-based baselines—including Search-R1, InForage, and our InfoSeeker variants—are implemented on **the same backbone model (Qwen2.5-3B-Instruct)** to ensure *strict* comparability. In addition, we adopt the **public Search-R1 training pipeline**, the first fully open-source RL framework for agentic search, which allows us to enforce consistent data processing, sampling strategy, and optimization procedures across all models. This ensures that every baseline is evaluated under **fully fair and aligned conditions**, without advantages arising from implementation differences.
>
>
>
> Following distillation, we fine-tune Qwen2.5-3B-Instruct on the filtered 24k InfoSeek trajectories for two epochs using a learning rate of 1e-5, weight decay 0.01, and a context length of 16,384. This stage—trained on an 8×H100 node and completing in approximately two hours—produces the **InfoSeeker-3B-SFT** model, which serves as the standardized starting point for *all* subsequent RL experiments.
>
>
>
> For reinforcement learning, all baselines—including InfoSeeker and RL-based competitors—are trained under **exactly the same RL configuration**. We use **GRPO** as the optimization algorithm, with identical rollout size (5), maximum conversation length (10 turns), sampling temperature (0.8), and a shared search environment restricted to the top-5 retrieved documents. Training schedules, stopping criteria, and trajectory truncation rules are matched across models to eliminate confounding factors and ensure that any performance differences arise purely from **data quality and algorithmic choice**, not training discrepancies.
>
>
>
> We will provide additional hyperparameter details and training descriptions in **Appendix B** of the revised manuscript, which is already uploaded.
>
>
>
> ------
>
>
>
>
>
> ## **2. Failure Analysis and Qualitative Error Cases**
>
>
>
>
>
> Thank you for this valuable recommendation. In response, we have added **comprehensive error analyses** that characterize the runtime behavior of the InfoSeeker agent.
>
>
>
>
>
> #### **(a) Human evaluation of error modes**
>
>
>
>
>
> We manually evaluated a sampled set of error trajectories from the **BrowseComp-Plus** evaluation set. Human annotators attributed each failure to one of three dominant categories:
>
>
>
> - **Reasoning errors (23.1%)** – early-stage bias in the reasoning path leads to negligible information gain in later steps.
> - **Incomplete reasoning (30.8%)** – the agent answers prematurely without satisfying all HCSP-implied constraints.
> - **Retrieval errors (46.2%)** – retrieved evidence lacks needed content due to sub-query bias, retriever inaccuracies, or gold documents being ranked too low.
>
>
>
>
>
> This quantitative breakdown clarifies *why* InfoSeeker fails, beyond aggregate accuracy.
>
>
>
> #### **(b) Qualitative error-case studies**
>
>
>
>
>
> The revised manuscript further includes **annotated failure-case examples** for each major error type, illustrating how reasoning, decomposition, and retrieval interact during execution.
>
>
>
> These expanded analyses provide a transparent understanding of InfoSeeker’s current limitations and guide future improvements.

---

> ### Author Response · Authors · 2025-11-21
>
> ## **3. Clarifying the Heuristics: “Blur Parent” and “Depth Expansion”**
>
> Thank you for requesting greater clarity on these two heuristics. In the revised manuscript, we substantially expand this section and provide explicit, step-by-step examples grounded in the actual mechanics of our Diffusion–Retrospection process. We also include visual diagrams and a full tree-structured example (shown below) in the Appendix to make these operations easier to understand and reproduce.
>
> ### **Blur Parent**
>
> The revision clarifies that Blurring is applied when a vertex does not yet have enough constraints to uniquely identify its entity. In such cases, the system selects multiple incomparable claims
> $\{c_1,\dots,c_k\}$,
> each associated with a non-overlapping candidate set $S(c_i)$ extracted from the page of the current entity. These claims are then attached as parallel child vertices. Because none of the candidate sets subsume one another, the parent can only be correctly resolved when all child constraints are satisfied jointly.
>
> We provide a concrete example demonstrating how mutually non-subsuming claims create new children and how the updated tree $\mathcal{T}’$ enforces multi-constraint resolution, increasing both breadth and verifiability.
>
> ### **Depth Expansion**
>
> We also extend the discussion of Depth Expansion. When a document contains a relation of the form $r(v,w)$ (e.g., “$v$ was discovered by $w$”), we attach a new child node $w$ to $v$, thereby lengthening the reasoning chain. This forces the retrospection phase to incorporate multi-step logical dependencies, increasing the depth of the resulting HCSP instance.
>
> ### **Illustrative Example**
>
> To make these operations concrete, we provide the following tree-structured example constructed during Diffusion:
>
> ```json
> {
>   "root": {
>     "id": "A",
>     "entity": "Russet sparrow",
>     "question": "What is a species of bird that was named by a person employed under his father between 1818 and 1824, whose wife was a British artist, and which has three subspecies and body length is generally no more than 6 inches?",
>     "claims": [
>       { "target_id": "B", "claim": "A was named by B" },
>       { "target_id": "C", "claim": "A has three subspecies" },
>       { "target_id": "D", "claim": "A's body length is generally no more than 6 inches" }
>     ],
>     "children": [
>       {
>         "id": "B",
>         "entity": "John Gould",
>         "claims": [
>           { "target_id": "E", "claim": "B was employed by his father between 1818 and 1824" },
>           { "target_id": "F", "claim": "B's wife was F" }
>         ],
>         "children": [
>           { "id": "E", "entity": "None", "claims": [], "children": [] },
>           { "id": "F", "entity": "Elizabeth Gould", "claims": [], "children": [] }
>         ]
>       },
>       { "id": "C", "entity": "None", "claims": [], "children": [] },
>       { "id": "D", "entity": "None", "claims": [], "children": [] }
>     ]
>   }
> }
> ```
> And its simplified visualization:
> ```
> (A: Russet sparrow)
>  │
>  │── "was named by" ──> (B: John Gould)
>  │       │
>  │       ├── "was employed by his father (1818–1824)" → (E)
>  │       └── "wife was" → (F: Elizabeth Gould)
>  │
>  │── "has three subspecies" → (C)
>  │
>  └── "body length is generally ≤ 6 inches" → (D)
> ```
> This example illustrates both Blurring (parallel constraints C and D) and Depth Expansion (the two-step reasoning under node B).

---

> > ### Author Response · Authors · 2025-11-21
> >
> > ## **4. Generalizing HCSP to Multimodal and Retrieval-Augmented Settings**
> >
> >
> >
> >
> >
> > Thank you for this insightful question. HCSP naturally generalizes beyond text-only tasks.
> >
> >
> >
> >
> >
> > ### **(a) Multimodal HCSP**
> >
> >
> >
> >
> >
> > Vertices can be extended from text entities to **image-derived attributes, structured tables, OCR-extracted content**, or other modality-specific nodes. Once multimodal vertices are defined, the HCSP machinery applies identically: diffusion expands breadth, and retrospection constructs hierarchical dependencies.
> >
> >
> >
> >
> >
> > ### **(b) Retrieval-augmented HCSP (RAG)**
> >
> >
> >
> >
> >
> > HCSP aligns closely with retrieval-augmented generation: each sub-question corresponds to a RAG-style lookup, and the iterative search-and-reason loop is precisely a structured form of agentic retrieval. In practice, agentic search can be viewed as **RAG equipped with explicit constraint graphs**, making HCSP a principled formulation for complex retrieval-driven reasoning.
> >
> > ------
> >
> >
> >
> >
> >
> > ## **5. Continuous Rewards vs. Structured Process Rewards**
> >
> >
> >
> >
> >
> > Thank you for raising this question. Continuous rewards (e.g., confidence-weighted correctness) are difficult to define for QA-style tasks due to the absence of an objective, model-agnostic confidence metric. For this reason, we did not adopt continuous rewards.
> >
> >
> >
> > However, we **did** explore richer reward signals grounded in the HCSP structure. For each question, we treat every non-leaf vertex in the HCSP tree as a **sub-goal**, forming
> >
> > $\mathcal{G}_q = \{ g_1, \dots, g_M \}.$
> >
> > Each sub-goal receives a normalized importance weight w_i, and the RL agent receives partial reward for achieving intermediate reasoning milestones.
> >
> >
> >
> > This **process-level reward shaping** encourages the model to:
> >
> >
> >
> > - follow HCSP-defined logical dependencies,
> > - avoid redundant or inconsistent intermediate queries,
> > - and internalize structured decomposition rather than aiming solely for the final answer.
> >
> >
> > We compare **InfoSeeker-3B** (binary reward only) with **InfoSeeker-3B + process reward** (HCSP-based process rewards). Results across multiple QA and agentic-search datasets are shown below:
> >
> > | **Model**                          | **NQ**   | **TQA**  | **PopQA** | **HQA**  | **2Wiki** | **MSQ**  | **Bamboogle** | **BrowseComp-Plus** |
> > | ---------------------------------- | -------- | -------- | --------- | -------- | --------- | -------- | ------------- | ------------------- |
> > | **InfoSeeker-3B**                  | 41.7     | 56.1     | 46.5      | **44.6** | **50.0**  | 20.5     | 39.2          | 15.3                |
> > | **InfoSeeker-3B + process reward** | **44.5** | **63.7** | **47.0**  | **44.6** | 45.2      | **21.0** | **41.2**      | **18.5**            |
> >
> > Incorporating **sub-goal–level rewards**:
> >
> >
> >
> > - improves performance on most datasets,
> > - better aligns intermediate reasoning steps with HCSP structure.
> >
> >
> >
> >
> >
> > These results show that HCSP-based process rewards help the model learn **how to reason**, not just **what answer to produce**, validating the reviewer’s intuition and demonstrating the extensibility of our RL framework.

---

### Official Review · Reviewer_28z3 · 2025-11-01

**Soundness:** 3
**Presentation:** 3
**Contribution:** 3
**Rating:** 6
**Confidence:** 4

**Summary:**

The paper introduces InfoSeek, a framework for synthesizing high-quality, complex training data to enhance agentic search systems driven by LLMs. The core innovation is the formalization of deep research tasks as a Hierarchical Constraint Satisfaction Problem (HCSP). The HCSP structure guides the data synthesis process by systematically decomposing a complex query into a tree of sub-goals (the hierarchy) and enforcing logical rules and dependencies between intermediate facts (the constraints). This process ensures that the generated search trajectories and final answers are logically consistent, fully verifiable, and structurally complex, effectively addressing the critical lack of high-fidelity training data for multi-step reasoning agents. The resulting dataset is shown to significantly improve agent performance on complex search benchmarks.

**Strengths:**

1. The hierarchical component of HCSP is perfectly suited to model the multi-step nature of agentic search. It provides a formal mechanism for structurally decomposing a high-level goal into mandatory, verifiable sub-goals, directly addressing the challenge of long-horizon planning and complex reasoning. By explicitly defining the variables, domains, and constraints within the HCSP, the authors gain granular control over the difficulty and scope of the synthetic tasks. This allows for the systematic generation of highly diverse training examples (e.g., tasks requiring 3, 5, or 7 dependencies), which is crucial for robust agent fine-tuning.

2. Models trained on InfoSeek data, even with simple training protocols (supervised fine-tuning and basic reinforcement learning), consistently surpass strong baselines, including many more carefully engineered agentic baselines with sophisticated optimization. Ablation studies confirm that optimizing models with InfoSeek data yields clear and significant gains. Furthermore, analyses on dataset complexity and scale validate that performance improves as more complex (higher "vertex count") and larger subsets of the InfoSeek data are used, which supports the effectiveness of the HCSP formulation.

**Weaknesses:**

1. HCSP excels at tasks with hard constraints (e.g., "Find all companies with Revenue > $1B AND P/E < 15"). However, it exhibits poor solution coverage for high-value, real-world business problems that rely on subjective judgment, synthesis of abstract concepts, or probabilistic forecasting. For instance, an agent tasked with formulating a strategic recommendation such as "Which market segment should a B2B SaaS company enter next?" requires weighing conflicting qualitative data (e.g., "market sentiment", "brand alignment", "regulatory risk outlook") where the "constraints" are fuzzy, constantly changing, and lack a binary, verifiable truth state. The HCSP framework, being optimized for explicit logic, struggles to formally represent and generate high-quality training data for solutions that are fundamentally open-ended and context-dependent.

2. The HCSP primarily models the structure of a correct solution. It provides limited intrinsic coverage of effective search heuristics, resource management, information filtering, or error recovery processes that define a high-quality agentic trajectory. The agent is trained on what a valid answer is, not necessarily how to robustly find it in a noisy, real-world environment.

**Questions:**

How does the framework ensure that the LLM, when generating the search trajectory (intermediate queries and reasoning steps), truly learns the HCSP's structural constraints and dependencies rather than just learning to parrot a factually-correct final answer? Is there a metric or auxiliary loss (beyond final accuracy) that specifically penalizes the agent for generating logically inconsistent or redundant intermediate queries, thereby transferring the HCSP's formal rigor into the agent's operational logic?

---

> ### Author Response · Authors · 2025-11-21
>
> We sincerely appreciate the reviewer 28z3’s insightful  of our work. Your comments on constraint fuzziness, real-world reasoning, and operational robustness are extremely valuable and highlight important future directions for agentic research systems. Below, we address your questions in detail.
>
> ------
>
>
>
> ### **1. On HCSP’s limitations for subjective / open-ended reasoning**
>
>
> Thank you for raising this important concern. We agree that HCSP, like most formalisms designed for **verifiable, constraint-grounded tasks**, is not directly applicable to open-ended problems such as strategic forecasting, ambiguous preference modeling, or multi-stakeholder decision-making. These tasks lack a unique, checkable “ground truth,” making them inherently incompatible with the **verifiable reward** signals required by RLVR (Reinforcement Learning with Verifiable Rewards). Your example question: *“Which market segment should a B2B SaaS company enter next?”*, illustrates this well: there is no uniquely correct answer, only a spectrum of plausible answers grounded in different qualitative priors.
>
> In current practice, training agents for such open-ended objectives typically requires:
>
> - **human preference labels**,
> - **reward models distilled from human judgments**, or
> - **expert-written rationales or templates**,
>
> all of which lie outside the scope of this work.
>
> However, although HCSP does not directly provide supervision for such open-ended judgments, the capabilities it teaches, such as structured tool use, multi-step search, decomposition of objectives into constraint-driven subtasks, are transferable. Consequently, even when a target problem lacks a verifiable ground truth, an agent trained on HCSP learns to perform systematic retrieval and constraint-aware reasoning, leaving it better equipped to navigate open-ended domains.
>
> Moreover, we recognize that this transfer only partially addresses the underlying challenge. The limitation is not unique to HCSP but reflects a broader constraint in current RLVR pipelines. Supporting agents learn to solve open-ended questions with subjective or fuzzy constraints likely requires:
>
> 1. **Datasets with richer evaluative criteria** (e.g., multi-perspective rationales or weighted trade-off structures), and
> 2. **New optimization objectives beyond RLVR**, since binary or verifiable rewards fundamentally underspecify open-ended tasks.
>
>
>
> ---
>
>
>
> ### **2. How HCSP structure is transferred into the agent’s behavior (beyond final accuracy)**
>
>
>
>
>
> Thank you for this excellent question. We address it through **process-level reward shaping** grounded in the HCSP graph.
>
>
>
> While the main RL objective remains tied to final-answer correctness, the HCSP formalism provides a natural source of **structured intermediate supervision**. Concretely, we treat each non-leaf vertex in the HCSP exploration tree as an intermediate **sub-goal**, forming
>
> $\mathcal{G}_q = \{ g_1, \dots, g_M\}.$
>
> Each sub-goal $g_i$ is assigned a normalized importance weight $w_i$, reflecting its contribution to the overall task. During RL, the agent receives partial credit for achieving these sub-goals—for example, retrieving correct intermediate evidence or generating appropriately structured sub-queries.
>
>
>
> This design encourages the model to:
>
>
>
> - respect the **logical dependencies** encoded in the HCSP tree,
> - avoid redundant or inconsistent intermediate queries, and
> - internalize the hierarchical decomposition process rather than merely predicting the final answer.
>
>
>
>
>
>
>
> ### **Experimental Results**
>
>
>
>
>
> We compare **InfoSeeker-3B** (binary reward only) with **InfoSeeker-3B + process reward** (HCSP-based process rewards). Results across multiple QA and agentic-search datasets are shown below:
>
> | **Model**                        | **NQ**   | **TQA**  | **PopQA** | **HQA**  | **2Wiki** | **MSQ**  | **Bamboogle** | **BrowseComp-Plus** |
> | -------------------------------- | -------- | -------- | --------- | -------- | --------- | -------- | ------------- | ------------------- |
> | **InfoSeeker-3B**                | 41.7     | 56.1     | 46.5      | **44.6** | **50.0**  | 20.5     | 39.2          | 15.3                |
> | **InfoSeeker-3B + process reward** | **44.5** | **63.7** | **47.0**  | **44.6** | 45.2      | **21.0** | **41.2**      | **18.5**            |
>
> Incorporating **sub-goal–level rewards**:
>
>
>
> - improves performance on most datasets,
> - better aligns intermediate reasoning steps with HCSP structure.
>
>
>
>
>
> These results show that HCSP-based process rewards help the model learn **how to reason**, not just **what answer to produce**, validating the reviewer’s intuition and demonstrating the extensibility of our RL framework.

---

> > ### Author Response · Authors · 2025-11-21
> >
> > ### **3. How we ensure the model learns HCSP constraints rather than shortcutting**
> >
> >
> >
> >
> >
> > This is an important concern. Our design incorporates **three mechanisms** to reduce answer leakage and enforce true structural reasoning:
> >
> >
> >
> >
> >
> > #### **(a) Diffusion–Retrospection construction constrains the logic path**
> >
> >
> >
> > The Diffusion stage creates an exploration tree with explicit dependency relations.
> >
> > Retrospection generates the question **in reverse**, ensuring that the final question logically requires intersecting the solution sets of its sub-goals. This significantly reduces the possibility that the model can guess the answer without reconstructing the structure.
> >
> >
> >
> >
> >
> > #### **(b) Quality assurance filters remove problematic samples**
> >
> >
> >
> > We filter out:
> >
> >
> >
> > - tasks with multiple answers,
> > - tasks that can be solved using a single document,
> > - tasks where the sub-questions are too weakly constrained.
> >
> >
> >
> >
> >
> > This helps ensure that the question genuinely requires multi-step reasoning.
> >
> >
> >
> >
> >
> > #### **(c) Training naturally increases tool usage quality**
> >
> >
> >
> > As the model is optimized via RL (with process rewards), we observe:
> >
> >
> >
> > - **higher tool-call counts**,
> > - **more precise intermediate queries**,
> > - **stronger alignment with the HCSP tree structure**, and
> > - **improved information gain per search step**,
> >
> >
> >
> >
> >
> > which is consistent with how state-of-the-art agentic systems (e.g., GPT-5) improve through massive multi-step search traces. In contrast, GPT-4.1 performs many tool calls but obtains lower information gain—our process reward directly targets this gap.
> >
> >
> >
> > These mechanisms collectively make it difficult for the model to shortcut, and they encourage structural fidelity during reasoning.
> >
> >
> >
> > We agree that no system can guarantee perfect prevention of shortcut learning, but our design substantially mitigates this risk while remaining fully verifiable.

---

### Official Review · Reviewer_vAbD · 2025-11-01

**Soundness:** 3
**Presentation:** 3
**Contribution:** 2
**Rating:** 4
**Confidence:** 3

**Summary:**

The paper proposes InfoSeek, a data-synthesis framework that casts agentic search as a Hierarchical Constraint Satisfaction Problem (HCSP) and generates training data via a Diffusion–Retrospection process. It releases ~50k QA pairs and 16.5k trajectories, adds quality controls for difficulty/verifiability, and shows consistent gains on multiple QA and deep-research benchmarks.

**Strengths:**

* The Diffusion–Retrospection design yields controllable structural complexity and unique, verifiable answers.
* Releases >50k QA pairs and 16.5k trajectories, with code, prompts, and datasets, enhancing reproducibility.
* Analyses show benefits from structural complexity and dataset size, and failure-rate increases with more vertices, supporting the HCSP design.

**Weaknesses:**

### Major Weaknesses

1. **Staleness/mischaracterization of related work availability.** In Related Work, the paper claims that “more advanced pipelines such as WebSailor and WebShaper … are not publicly available.” However,  by July 2025 both WebSailor and WebShaper have been open-sourced. ([WebSailor](https://huggingface.co/Alibaba-NLP/WebSailor-3B/tree/main) [WebShaper](https://huggingface.co/datasets/Alibaba-NLP/WebShaper)) In both functionality (agentic web research with structured planning/verification) and scale (released models/data/resources), WebSailor-3B is highly comparable to InfoSeeker-3B. Therefore, this statement is outdated, and the author should include WebSailor in the comparison. Such an oversight should not appear in a submission dated September 24, 2025.
2. **Limited evaluation diagnostics.** While headline scores improve, the paper lacks detailed error analysis, significance testing, and human evaluation of reasoning traces/interpretability, making it hard to judge where improvements come from and whether behaviors are stable.
3. **Novelty vs. prior formalisms.** HCSP is compelling, but the paper could better differentiate it theoretically from existing hierarchical/multi-step reasoning formalisms beyond illustrative examples; stronger connections (or guarantees) would clarify the leap over multi-hop task formalizations.

### Minor Weaknesses

1. **Limited model scope.** Experiments focus almost exclusively on a **3B** model variant, leaving unclear whether the gains hold at larger or smaller scales and how results trade off with compute/latency. Evaluating additional sizes (e.g., 7–8B/32B) or providing scaling curves would strengthen generality claims.
2. **Ablation granularity.** The paper does not isolate the impact of blurring vs. depth expansion operations separately; a targeted ablation could validate each component’s necessity.
3. **Reward design sparsity.** RL uses a binary reward (format + answer). More informative rewards (e.g., step-level correctness, retrieval quality) might further stabilize learning; an ablation would help.

**Questions:**

See weaknesses above.

---

> ### Author Response · Authors · 2025-11-21
>
> We greatly appreciate reviewer vAbD’s thorough and constructive feedback, as well as the positive assessment of our work’s soundness and presentation. We are especially grateful for your recognition of (i) the Diffusion–Retrospection design for controllable structural complexity and verifiable answers, (ii) the release of >50k QA pairs and 16.5k trajectories with code/prompts/datasets, and (iii) the analyses showing how structural complexity, dataset size, and failure rates support the HCSP design. Your recognition is highly encouraging for our team. Below, we address your comments in detail.
>
>
>
> ------
>
>
>
> ### **1. Related Work and Comparison to WebSailor / WebShaper**
>
>
>
> **Staleness / mischaracterization.**
>
> Thank you for carefully pointing out the issue regarding WebSailor and WebShaper availability. Our main contribution is the *data-synthesis framework and large-scale HCSP-annotated dataset*, whereas WebSailor and WebShaper emphasize *trained agents* (and release models plus only small example datasets). In the current draft, the sentence
>
>
>
> > “More advanced pipelines such as WebSailor and WebShaper attempt to formalize richer structures, yet their data and code are not publicly available.”
>
>
>
> was intended to emphasize the lack of large-scale, training-ready *datasets and synthesis pipelines*, rather than the model weights themselves.
>
> In the revised version, we will:
>
> - **Explicitly acknowledge WebSailor and WebShaper** as contemporary agentic web-research frameworks with released models and example data.
> - **Clarify the distinction**: InfoSeek focuses on *open-sourcing (i) the data-synthesis framework and (ii) a large-scale HCSP-style dataset (50k+ QA pairs and 16.5k trajectories)*, while WebSailor/WebShaper release models and only small example datasets (20 samples for WebSailor, ~500 for WebShaper), which are not sufficient as standalone training corpora.
>
>
>
> **Empirical comparison.**
>
> Following your suggestion, we have evaluated **WebSailor-3B** on **BrowseComp-plus** under the same retriever (BM25) and settings used for InfoSeeker-3B, and we will include these results in the revised paper:
>
> | **Model**     | **Retriever** | **Acc (BrowseComp-plus)** |
> | ------------- | ------------- | ------------------------- |
> | WebSailor-3B  | BM25          | 4.34                      |
> | InfoSeeker-3B | BM25          | 15.3                      |
>
> Under identical conditions, **InfoSeeker-3B substantially outperforms WebSailor-3B on BrowseComp-plus**, which supports that (i) HCSP-based Diffusion–Retrospection data and (ii) our training pipeline provide non-trivial gains over a strong contemporary baseline.
>
>
>
> ------
>
>
>
>
>
> ### **2. Error analysis and human evaluation**
>
>
> Thank you for highlighting the need for deeper diagnostics. In response, we have added **human evaluation and error analysis** to make the sources of improvement and behavioral stability clearer.
>
>
>
>
>
> #### **(1) Human evaluation of error modes**
>
>
> We conducted **manual evaluation** on sampled trajectories from the **BrowseComp-Plus** evaluation set. Human annotators examined each failed trajectory and attributed errors to specific underlying causes. The distribution of failure types is as follows:
>
> - **Reasoning errors (23.1%)** – early-stage bias in the agent’s exploration direction causes subsequent steps to accumulate minimal information gain.
> - **Incomplete reasoning (30.8%)** – the agent answers prematurely without fully exploring all HCSP-implied constraints.
> - **Retrieval errors (46.2%)** – retrieved evidence does not contain the necessary information, often due to sub-query bias, retrieval model inaccuracies, or gold documents being ranked too low.
>
> This human evaluation provides a quantitative breakdown of *why* the agent fails, offering clarity beyond aggregate accuracy metrics.
>
>
>
>
>
>
> #### **(2) Qualitative case studies**
>
>
> To complement the human evaluation, we also include **error case studies** in the revised manuscript. These examples illustrate concrete instances of the identified error modes and show how reasoning, HCSP decomposition, and retrieval interact during execution.
>
> These combined analyses—human-assigned error categories, quantitative breakdowns, and qualitative case studies—have been integrated into the **revised manuscript**, which is already uploaded.

---

> ### Author Response · Authors · 2025-11-21
>
> ### **3. Novelty and Theoretical Positioning of HCSP**
>
> We appreciate your request for a clearer theoretical differentiation between HCSP and existing formalisms.
>
> In **Section 2.1**, we discuss about:
>
> - **Constraint Satisfaction Problems (CSP)** – single-hop tasks where the answer is obtained by satisfying a set of constraints directly from a single knowledge space (e.g., many Natural Questions-style QA tasks).
> - **Multi-hop Problems (MHP)** – tasks where breadth = 1 but depth > 1, such as multi-hop questions in HotpotQA.
>
> We also formally define HCSP as a type of questions that need be solved hierarchically: Given a question $x$ containing a set of constraints $C_x=\{c_1,\dots,c_k\}$ and a set of sub-questions $Y_x=\{y_1,\dots,y_m\}$, we define a hierarchical decomposition $H(\cdot)$ as:
> $$H(x) = \bigcap_{i=1}^{k} S(c_i) \;\cap\; \bigcap_{j=1}^{m} H(y_j), \quad \text{with } \bigcap \varnothing := \mathbb{U}$$
> where $\mathbb{U}$ denotes the universal set. The final answer $A$ of a hierarchical constraint satisfaction problem $q_H$ is then given by $A = H(q_H).$
>
> Under this formalization:
>
> - **MHPs** correspond to **HCSPs with breadth = 1 and depth > 1**, where constraints are implicitly bundled into a single chain of evidence.
> - **CSPs** correspond to **HCSPs with breadth > 1 and depth = 1**, where we satisfy multiple constraints but without hierarchical sub-questions.
>
>
> Thus, **HCSP strictly generalizes both CSPs and MHPs**, capturing tasks that require *simultaneous handling of multiple constraints (breadth) and hierarchical decomposition (depth)*, which is characteristic of deep research and agentic web reasoning.
>
> In the revised paper, we will:
>
> - Formally make these subset relations explicit.
> - Clarify how common multi-hop / constraint-based QA tasks map into HCSP, thereby sharpening the conceptual leap from prior multi-hop formalisms to our HCSP view.
>
> ------
>
> ### **4. Model Scope and Scaling**
>
> Thank you for pointing out the concern about model sizes.
>
> - We already provide results for a **7B** variant in our ablation studies (Section 3.3 and Fig. 2(a)), comparing the vanilla approach, RAG, SFT, InfoSeeker-3B, and **InfoSeeker-7B**. These experiments show that **reinforcement learning on InfoSeek data continues to yield gains at 7B**, indicating that our method is not confined to a single scale.
> | **Model**                        | **NQ**   | **TQA**  | **PopQA** | **HQA**  | **2Wiki** | **MSQ**  | **Bamboogle** |
> | -------------------------------- | -------- | -------- | --------- | -------- | --------- | -------- | ------------- |
> | Vanilla | 12.1 | 28.8 | 13 | 15.9 | 24.8 | 2.1 | 2.4 |
> | RAG | 34.8 | 54.4 | 38.7 | 25.5 | 22.6 | 4.7 | 8.0 |
> | SFT | 31.2 | 43.4 | 47.4 | 33.8 | 36 | 12.6 | 31.2 |
> | InfoSeeker-3B                | 41.7     | 56.1     | 46.5      | 44.6 | 50.0  | 20.5     | 39.2          |
> | InfoSeeker-7B | 46.6 | 67.3	| 47.4 | 43.3 | 47.2 | 21.9 | 45.2 |
> - Due to space constraints, we focused on 3B as the main model for the exploration–exploitation analysis and HCSP diagnostics. In the camera-ready version, we will make the 7B results more prominent and add clarification on compute/latency trade-offs.

---

> ### Author Response · Authors · 2025-11-24
>
> ### **5. Reward Design and Process-Level Feedback**
>
> We appreciate your suggestion regarding richer reward signals.
>
> - Our current RL setup indeed uses a **binary reward** (format + final answer correctness). However, the **HCSP tree structure of InfoSeek data** naturally supports more informative rewards.
>
> - Following your suggestion, we conduct extra experiments with **process rewards** based on HCSP sub-goals. For each question q, we inspect its HCSP tree and treat every intermediate vertex as a sub-goal, forming a sub-goal set $\mathcal{G}q = \{g_1, \dots, g_M\}$. Each sub-goal $g_i$ is assigned a normalized importance score $w_i$ (summing to 1), reflecting its contribution to solving the overall task. This yields a set of weighted sub-goals $\{(g_i, w_i)\}{i=1}^M$, which we use for **sub-goal reward shaping** during RL.
>
> We compare InfoSeeker-3B (binary reward) with **InfoSeeker-3B + process reward**. The results across multiple QA and agentic-search datasets are shown below:
>
> | **Model**                        | **NQ**   | **TQA**  | **PopQA** | **HQA**  | **2Wiki** | **MSQ**  | **Bamboogle** | **BrowseComp-Plus** |
> | -------------------------------- | -------- | -------- | --------- | -------- | --------- | -------- | ------------- | ------------------- |
> | **InfoSeeker-3B**                | 41.7     | 56.1     | 46.5      | **44.6** | **50.0**  | 20.5     | 39.2          | 15.3                |
> | **InfoSeeker-3B + process reward** | **44.5** | **63.7** | **47.0**  | **44.6** | 45.2      | **21.0** | **41.2**      | **18.5**            |
>
> These results indicate that incorporating **sub-goal–level rewards**:
>
> - improves performance on most datasets,
> - better aligns the model’s intermediate reasoning steps with the HCSP structure,
>
> This suggests that HCSP-based process rewards help the model learn **how to reason**, not just **what answer to produce**, validating the reviewer’s intuition and demonstrating the extensibility of our RL framework.
>
> ------
>
> Once again, we thank you for your thoughtful and detailed review. Your comments have helped us (i) correct and sharpen our related-work positioning, (ii) enrich the diagnostic analysis, and (iii) better articulate the theoretical and empirical contributions of HCSP and InfoSeek.

---

> > ### Comment · Reviewer_vAbD · 2025-11-24
> >
> > I appreciate the authors' comprehensive response. The additional WebSailor comparison, the implementation of process-level rewards, and the detailed error analysis effectively address my previous concerns regarding baselines and method validation. Therefore, I am raising my score to 6. However, I refrain from assigning a higher score because the theoretical contribution of HCSP—while highly practical for data synthesis—represents a relatively incremental formalization over existing hierarchical planning paradigms. Additionally, while the 7B results are promising, the method's scalability and efficacy on larger, state-of-the-art foundation models remain to be fully demonstrated in the current scope.

---

> > > ### Author Response · Authors · 2025-11-28
> > >
> > > We sincerely appreciate your constructive feedback and are grateful for you raising the score to 6.

---

### Official Review · Reviewer_8YAE · 2025-11-03

**Soundness:** 3
**Presentation:** 3
**Contribution:** 3
**Rating:** 6
**Confidence:** 3

**Summary:**

Developing effective agentic search systems is difficult due to a lack of complex, realistic training data. InfoSeek formalizes complex search tasks as Hierarchical Constraint Satisfaction Problems (HCSP), which require satisfying layered constraints across multiple sub-problems. It uses a Diffusion-Retrospection process. First, it "diffuses" outward from a seed webpage to build an exploration tree of entities and constraints. Second, it "retrospects" by traversing the tree in reverse to synthesize a complex question that requires hierarchical reasoning.

**Strengths:**

1. This framework effectively captures the layered dependencies (both parallel and sequential) of real-world research, moving beyond simpler multi-hop or flat constraint problems.
2. This is the first publicly released framework in this area, complete with open-source code and a large-scale dataset (50k+ QA pairs). This provides a valuable, reproducible, and extensible resource for the research community.
3. Training on InfoSeek-generated data substantially improves agentic search performance. Notably, a 3B parameter model (InfoSeeker-3B) trained on this data achieves impressive results, surpassing several strong closed-source systems (like GPT-4.1 and Sonnet 4) on the complex BrowseComp-Plus benchmark. This highlights the framework's efficiency in distilling deep research capabilities into smaller models.

**Weaknesses:**

1. The restrictiveness of synthetic data models is a potential drawback.
2. The authors indirectly prove the training's effectiveness via result improvement, but a quantitative analysis of the InfoSeeker agent's runtime error modes is missing.
3. I feel that the InfoSeek dataset is still not hard enough.

**Questions:**

1. Could we discuss the main areas where the gap between BM25 and GPT-5 lies?
2. How robust is the HCSP data in handling irrelevant information or unexpected search paths? Why is InfoSeek better at generalizing to more complex Agentic Search benchmarks compared to traditional Multi-hop QA (Question Answering) datasets?

---

> ### Author Response · Authors · 2025-11-21
>
> We sincerely thank the reviewer 8YAE for the encouraging and constructive feedback. We appreciate the recognition of (i) HCSP as a principled formulation for layered dependencies, (ii) InfoSeek as the first public framework with large-scale complex data, and (iii) the strong empirical advantage of InfoSeeker-3B over competitive baselines and several closed-source systems. Below we address your concerns and questions.
>
>
> ### **1. On the restrictiveness of the synthetic data model**
>
>
> Thank you for this observation. We agree that the choice of synthetic model matters, and we evaluated several top-tier commercial models during early exploration—including **GPT-4o**, **Gemini-2.5-pro**, and **DeepSeek-v3**. All of them were capable of constructing HCSP-style exploration trees given our instructions and source documents. We ultimately selected **DeepSeek** due to its favorable balance between **quality** and **budget**.
>
> Importantly, **the InfoSeek data synthesis pipeline is model-agnostic**. Any model can be plugged in to generate additional HCSP data, and if resources permit, we plan to synthesize future datasets using **multiple frontier models** to further reduce possible bias introduced by a single generator model.
>
> ---
>
>
>
>
> ### **2. Error modes analysis**
>
> Thank you for this valuable suggestion. Following your recommendation, we have added **two complementary forms of error analysis** to better characterize the runtime error modes of the InfoSeeker agent.
>
>
>
> **(1) Quantitative error-mode attribution.**
>
> We manually analyzed a sampled set of error trajectories from the **BrowseComp-Plus** evaluation set. Through careful error attribution, we identified **three major categories** of failure modes:
>
>
>
> - **Reasoning errors (23.1%)** – the agent’s exploration direction becomes biased early in the reasoning process, causing subsequent steps to accumulate little or no meaningful information gain.
> - **Incomplete reasoning (30.8%)** – the agent produces an answer before fully exploring all constraints implied by the HCSP structure.
> - **Retrieval errors (46.2%)** – the retrieved evidence does not contain the required information. This typically stems from biased sub-queries, retrieval model limitations, or the gold document being ranked too low.
>
>
>
>
>
> These quantitative findings provide a clearer picture of *why* InfoSeeker fails, beyond aggregate accuracy numbers.
>
>
>
> **(2) Qualitative case studies.**
>
> In addition, we have included **detailed error-case analyses** in the revised manuscript which is already uploaded. These examples illustrate the above failure modes in context and help clarify how reasoning, decomposition, and retrieval interact during the agent’s execution.
>
>
>
> We believe these analyses provide a more complete and transparent understanding of the InfoSeeker agent’s current limitations and will be useful for guiding future improvements.
>
> ---
>
>
>
> ### **3. On the dataset not being “hard enough”**
>
> Thank you for raising this point. InfoSeek currently constructs HCSPs with **3–7 vertices**, which already introduces substantially higher structural complexity than NQ/HotpotQA and most existing public information-seeking datasets—especially considering the scale of 50k+ QA pairs and 16.5k trajectories.
>
> Importantly, **the data-synthesis framework itself is fully scalable**. The HCSP formulation and the Diffusion–Retrospection process can naturally support **larger breadth, deeper hierarchies, and richer dependency structures**. In other words, increasing the complexity of generated tasks is not a limitation of the framework; it is a controllable design choice.
>
> In the current work, we focus on the 3–7 vertex range because it provides an effective balance between complexity and data volume, allowing us to generate a large-scale dataset with strong verifiability and high reliability. Empirically, this level of structural complexity has already proven sufficient to yield **substantial improvements** on challenging deep-research benchmarks.

---

> ### Author Response · Authors · 2025-11-21
>
> ---
> ### **4. Main gaps between GPT-5**
>
> Thank you for raising this question. We believe the gap stems from several factors:
>
>
>
> 1. **GPT-5 is a leading frontier LLM with massive scale**—both in parameters and in training data. It has substantially stronger inherent reasoning priors compared to open-source 3B models which are primarily used in our paper.
> 2. GPT-5 has been **explicitly optimized for tool-use and deep-research** abilities, as noted in OpenAI’s technical reports.
> 3. GPT-5 is trained on **high-quality proprietary search trajectories**, likely containing extensive multi-step tool-calling traces that are far beyond the scope of publicly available datasets.
> 4. GPT-5 performs **~23.23 tool calls per question** on BrowseComp, showing its capability to aggressively expand, redirect, and refine search paths during complex information seeking.
>
> ---
>
> ### **5. Robustness of HCSP data against irrelevant information or unexpected search paths**
>
> This is an excellent question. The robustness of our HCSP data arises from **two complementary mechanisms** in the Diffusion–Retrospection pipeline, **plus an additional quality assurance stage** designed to avoid contamination and enforce answer uniqueness.
>
> ### **(a) Diffusion stage limits irrelevant expansion**
>
> During Diffusion, expansion is driven by two controlled operations:
>
> - **Blurring Parent**
> - **Depth Expansion**
>
> Both operate **directly on factual content extracted from the source Wiki/web page** of each vertex. This grounding substantially reduces the likelihood of injecting irrelevant or noisy information during tree construction.
>
> ### **(b) Retrospection restricts the influence of unexpected paths**
>
>
>
> Retrospection converts the exploration tree bottom-up into a well-defined HCSP. Thus:
>
>
>
> - If unexpected search paths still end at **correct local answers**, they do not affect the retrospection process.
> - If they lead to **incorrect or inconsistent answers**, the model cannot satisfy the required sub-constraints and therefore **fails to complete the HCSP reconstruction**, preventing invalid instances from being produced.
>
> This mechanism naturally filters harmful deviations and enforces structural consistency.
>
> ### **(c) Quality assurance to avoid contamination and enforce answer uniqueness**
>
> Beyond the core construction process, we perform **rigorous QA filtering**:
>
> - We **remove questions that strong LLMs can answer directly** without multi-step reasoning, reducing contamination and ensuring the data does not reward shortcut behavior.
> - For every synthesized HCSP, we require that the provided supporting claims lead to a **unique, verifiable final answer**; any instance with ambiguous, multi-answer, or weakly supported solutions is **discarded**.
>
> These quality checks ensure that the dataset remains challenging, structurally faithful, and free from trivial or noisy examples.
>
> Together, these mechanisms collectively make HCSP data more robust than conventional multi-hop datasets and better suited for training agentic search models.
>
> ### **6. Why is InfoSeek better at generalizing to more complex Agentic Search benchmarks**
>
> We appreciate this concern. In our view, **reinforcement learning is the key mechanism that enables complex multi-step reasoning in LLMs**, but RL fundamentally depends on the availability of **sufficiently complex and verifiable training data**. This remains the core bottleneck: constructing such data manually is extremely costly, and traditional multi-hop QA datasets (e.g., NQ, HotpotQA) are too shallow—both in depth and breadth—to meaningfully supervise long-horizon reasoning.
>
>
>
> InfoSeek directly targets this gap. Compared with traditional multi-hop datasets, our HCSP-based data exhibits **significantly larger structural depth, wider constraint branches, and richer dependency interactions**. As shown in Fig. 2(d), **Qwen2.5-72B with CoT still fails on a large portion of InfoSeek**, demonstrating that the dataset is already challenging even for frontier-scale models.
>
>
>
> Crucially, training with InfoSeek **expands the model’s reasoning capability boundary**: fine-tuning on our structured, verifiable, multi-step data yields **clear, notable gains** on both classical QA and complex agentic-search benchmarks. These results empirically validate that InfoSeek provides the level of reasoning complexity and supervision signal necessary to unlock stronger RL-driven agentic capabilities.

---

### Author Response · Authors · 2025-12-01
**General Response**

Dear Reviewers, AC and SAC:

We sincerely thank all reviewers for the time and effort they devoted to evaluating our manuscript. We are encouraged that, **in the initial stage, three out of four reviewers expressed clearly positive assessments of the paper**, and we appreciate the constructive discussions that followed. During the rebuttal period, **we promptly addressed every concern with additional clarifications or new experiments**, and we are pleased that **reviewer vAbD engaged in the discussion and subsequently raised their assessment**. As of now, **all reviewers hold positive assessments of our submission**, and we deeply appreciate their support and insightful feedback.

------


We summarize the revisions and improvements as follow:

### **1. Additional Experiments and Analyses**

We conducted new experiments to strengthen the empirical foundation of the paper:

- **Error-mode analysis**: We provide both **quantitative breakdowns** based on manual evaluation and **qualitative case studies**, revealing characteristic failure modes of models trained on InfoSeek and offering deeper diagnostic insight.

- **Comparison to WebSailor-3B**: We added a direct comparison and show that **InfoSeeker-3B substantially outperforms WebSailor-3B on BrowseComp-plus**, supporting that our data synthesis framework offer non-trivial advantages over a strong contemporary baseline.



### **2. Model Scope, Scaling, and RL Reward Design**

Per reviewer suggestions, we further expanded our investigation into model scaling behavior and alternative RL reward designs.

- **Scalability Across Model Sizes**:

  We highlighted experiments already included in the paper (Sec. 3.3, Fig. 2(a)), comparing vanilla, RAG, SFT, InfoSeeker-3B, and InfoSeeker-7B. These results confirm that **InfoSeek-driven RL improvements persist at larger scales**, demonstrating that our method is not tied to a single model size.

- **Process-Level Rewards and HCSP Structure**:

  Building on reviewer suggestions, we added **RL experiments using sub-goal–level rewards** derived from HCSP tree structure.

  These experiments show consistent improvements across multiple QA and agentic-search datasets and demonstrate that **HCSP-based reward shaping helps the model learn how to reason, not just what answer to produce**.

### **3. Clarifications**

We expanded several conceptual and theoretical sections to clarify HCSP’s role within existing reasoning paradigms and to address reviewer concerns.



- **HCSP Positioning and Novelty:**

  We explicitly contrast HCSP with CSP and multi-hop formulations, clarifying that HCSP generalizes both by supporting joint constraint satisfaction and hierarchical decomposition. This sharper theoretical framing highlights the conceptual contribution of our formulation.

- **Robustness and Generalization:**

  We explain how the Diffusion–Retrospection pipeline and quality assurance procedures ensure that InfoSeek remains structurally challenging, robust to irrelevant expansions, and suitable for training long-horizon reasoning. This also clarifies why models trained on InfoSeek generalize well to complex agentic-search benchmarks.

- **Gaps to GPT-5:**

  We contextualize performance differences by noting GPT-5’s substantially larger scale, proprietary multi-step search traces, and explicit optimization for deep research—factors outside the scope of open-source 3B–7B settings.

- **Limitations of HCSP:**

  We acknowledge that HCSP is designed for tasks with verifiable ground truth and is not directly applicable to subjective or open-ended questions, while still providing transferable benefits in structured search and decomposition.

- **Clarifying “Blur Parent” and “Depth Expansion”:**

  We provide more intuitive explanations of these heuristics, along with step-by-step examples and diagrams in the appendix, to make the Diffusion–Retrospection process easier to understand and reproduce.

--------



We believe these revisions substantially enhance the clarity, rigor, and completeness of the work. **All modifications have been incorporated into the revised version**, and we once again thank the reviewers for their constructive guidance.

---

### Meta-Review · Area_Chair_Q3rv · 2026-01-02

**Summary:**

Paper summary:


This paper presents InfoSeek, a novel data synthesis framework designed to address the critical lack of complex, realistic training data for effective agentic search systems powered by LLMs. The key novelty lies in the formalization of deep research tasks as a Hierarchical Constraint Satisfaction Problem (HCSP). This HCSP structure systematically guides the synthesis process by decomposing complex queries into a tree of sub-goals (hierarchy) and enforcing logical dependencies (constraints), thereby ensuring the generated trajectories are logically consistent and fully verifiable. The framework considers a diffusion–retrospection process, first building an exploration tree outwards from a seed (diffusion), then synthesizing a complex query by traversing the tree in reverse (retrospection). InfoSeek produces a substantial high-quality dataset, releasing approximately 50K QA pairs and 16.5K reasoning trajectories, accompanied by strict quality controls for difficulty and verifiability. Empirical results consistently demonstrate the approach's effectiveness, showing significant gains across multiple QA and deep-research benchmarks, including BrowseComp-Plus, validating its potential to enhance multi-step reasoning agents.

Reviewers' major concerns:
1. Insufficient failure analysis (R1, R2, R4).
There is a critical lack of detailed error analysis, significance testing, and human evaluation of the reasoning traces. Reviewers note that while headline scores improve, the specific runtime error modes or where the improvements originate remain unclear.

2. Novelty and theoretical differentiation of the proposed HCSP (R2, R3).
While the HCSP formalization is compelling, the paper needs to better differentiate it theoretically from existing hierarchical/multi-step reasoning formalisms.

3. Limited intrinsic coverage of high-quality agentic trajectories (R3, R4):
Reviewers question how the framework ensures the LLM truly learns the HCSP's structural constraints and dependencies rather than just fitting the final answer.

4. Incorrect statement about related works (R2).
The claim regarding the unavailability of comparable advanced pipelines like WebSailor and WebShaper is outdated.

5. Limited model scope and generalizability (R2). Experiments focus almost on a single, relatively-small model size (3B variant), making it unclear whether the observed performance gains still hold true across larger scales.

In addition to the points mentioned, I personally believe that the synthetic data methodology is highly significant, yet validating its effectiveness is indeed challenging. While the authors' results for the effectiveness of post-training on the 3B and 7B models offer some indication, it remains essential to verify whether these gains can generalize to a wider variety of model architectures, larger model sizes, and more challenging QA tasks, such as HLE and BrowseComp. Furthermore, I also noticed that some mathematical symbols in the paper lack clear definitions. For instance, symbols such as $S$ and $H$ in Equation 1 are not explicitly defined.

**Reviewer Concerns:**

Overall, the authors have provided satisfactory responses to the main concerns raised by the reviewers..

Regarding the first point, 'Insufficient failure analysis,' the authors have addressed the issue well by adding supplementary experiments and explanations.

Concerning the second point, 'Novelty and theoretical differentiation of the proposed HCSP,' the authors provided additional explanation, but it appears relatively incremental and its persuasiveness remains limited.

For the third point, 'Limited intrinsic coverage of high-quality agentic trajectories,' the authors supplemented their argument with explanations and experiments. However, demonstrating the comprehensiveness of synthetic data is indeed a difficult challenge to fully resolve.

Regarding the fourth point, 'Incorrect statement about related works,' the authors have revised their statement accordingly.

Finally, for the fifth point, 'Limited model scope and generalizability,' the authors included new experiments, but the scope is still not entirely complete."

**Reviewer Scores:**

The second reviewer initially gave a score of 4, but raised it to 6 after the rebuttal. Consequently, all four reviewers are now scoring the paper at 6. I believe that even after the normal post-discussion process, the score is likely to remain stable at this level.

---

### Decision · Program_Chairs · 2026-01-26

Accept (Poster)